# A chromosome-level genome of *Astyanax mexicanus* surface fish for comparing population-specific genetic differences contributing to trait evolution

Wesley C. Warren[1,2 ✉], Tyler E. Boggs[3], Richard Borowsky [4], Brian M. Carlson [5], Estephany Ferrufino[6], Joshua B. Gross[3], LaDeana Hillier[7], Zhilian Hu [8], Alex C. Keene[9], Alexander Kenzior[10], Johanna E. Kowalko[6], Chad Tomlinson [11], Milinn Kremitzki[11], Madeleine E. Lemieux [12], Tina Graves-Lindsay[11], Suzanne E. McGaugh[13], Jeffrey T. Miller [13], Mathilda T. M. Mommersteeg [8], Rachel L. Moran [13], Robert Peuß [10,18], Edward S. Rice[14], Misty R. Riddle[15,19], Itzel Sifuentes-Romero[6], Bethany A. Stanhope[6,9], Clifford J. Tabin[15], Sunishka Thakur[6], Yoshiyuki Yamamoto [16] & Nicolas Rohner [10,17 ✉]

Identifying the genetic factors that underlie complex traits is central to understanding the mechanistic underpinnings of evolution. Cave-dwelling *Astyanax mexicanus* populations are well adapted to subterranean life and many populations appear to have evolved troglo-morphic traits independently, while the surface-dwelling populations can be used as a proxy for the ancestral form. Here we present a high-resolution, chromosome-level surface fish genome, enabling the first genome-wide comparison between surface fish and cavefish populations. Using this resource, we performed quantitative trait locus (QTL) mapping analyses and found new candidate genes for eye loss such as *dusp26*. We used CRISPR gene editing in *A. mexicanus* to confirm the essential role of a gene within an eye size QTL, *rx3*, in eye formation. We also generated the first genome-wide evaluation of deletion variability across cavefish populations to gain insight into this potential source of cave adaptation. The surface fish genome reference now provides a more complete resource for comparative, functional and genetic studies of drastic trait differences within a species.

[1] Department of Animal Sciences, Institute for Data Science and Informatics, Bond Life Sciences Center, University of Missouri, Columbia, MO, USA.
[2] Department of Surgery, Institute for Data Science and Informatics, Bond Life Sciences Center, University of Missouri, Columbia, MO, USA. [3] Department of Biological Sciences, University of Cincinnati, Cincinnati, OH, USA. [4] Department of Biology, New York University, New York, NY, USA. [5] Department of Biological Sciences, Northern Kentucky University, Highland Heights, KY, USA. [6] Harriet L. Wilkes Honors College, Florida Atlantic University, Jupiter, FL, USA. [7] Department of Genome Sciences, University of Washington, Seattle, WA, USA. [8] Department of Physiology, Anatomy and Genetics, University of Oxford, Oxford, UK. [9] Department of Biological Sciences, Florida Atlantic University, Jupiter, FL, USA. [10] Stowers Institute for Medical Research, Kansas City, MO, USA. [11] McDonnell Genome Institute, Washington University, St Louis, MO, USA. [12] Bioinfo, Plantagenet, ON, Canada. [13] Department of Ecology, Evolution, and Behavior, University of Minnesota, Saint Paul, MN, USA. [14] Bond Life Sciences Center, University of Missouri, Columbia, MO, USA. [15] Genetics Department, Blavatnik Institute, Harvard Medical School, Boston, MA, USA. [16] Department of Cell and Developmental Biology, University College London, London, UK. [17] Department of Molecular & Integrative Physiology, KU Medical Center, Kansas City, KS, USA. [18]Present address: Institute for Evolution and Biodiversity, University of Münster, Münster, Germany. [19]Present address: Department of Biology, University of Nevada, Reno, NV, USA.
✉email: wwarren@genome.wustl.edu; nro@stowers.org

Establishing the relationship between natural environmental factors and the genetic basis of trait evolution has been challenging. The ecological shift from surface-to-cave environments provides a tractable system to address this, as in this instance the polarity of evolutionary change is known. Across the globe, subterranean animals, including fish, salamanders, rodents, and myriads of invertebrate species, have converged on reductions in metabolic rate, eye size, and pigmentation[1–3]. The robust phenotypic differences from surface relatives provide the opportunity to investigate the mechanistic underpinnings of evolution and to determine whether genotype–phenotype interactions are deeply conserved.

The Mexican cavefish, *Astyanax mexicanus*, has emerged as a powerful model to investigate complex trait evolution[4]. *A. mexicanus* comprises at least 30 cave-dwelling populations in the Sierra de El Abra, Sierra de Colmena, and Sierra de Guatemala regions of northeastern Mexico, and surface-dwelling fish of the same species inhabit rivers and lakes throughout Mexico and southern Texas[5]. The ecology of surface and cave environments differ dramatically, allowing for functional and genomic comparisons between populations that have evolved in distinct environments. Dozens of evolved trait differences have been identified in cavefish including changes in morphology, physiology, and behavior[4,6]. It is also worth noting that comparing cavefish to surface fish reveals substantial differences in many traits of possible relevance to human disease, including sleep duration, circadian rhythmicity, anxiety, aggression, heart regeneration, eye and retina development, craniofacial structure, insulin resistance, appetite, and obesity[7–17]. Further, generation of fertile surface-cave hybrids in a laboratory setting has allowed for genetic mapping in these fish[10,18–25]. Clear phenotypic differences, combined with availability of genetic tools, positions *A. mexicanus* as a natural model system for identifying the genetic basis of ecologically and evolutionary relevant phenotypes[17,26,27]. Recently, the genome of an individual from the Pachón cave population was sequenced[28]. While this work uncovered genomic intervals and candidate genes linked to cave traits, an important computational limitation was sequence fragmentation due to the use of short-read sequencing. In addition, lack of a surface fish genome prevents direct comparisons between surface and cavefish populations at the genomic level.

Here, we address these two obstacles by presenting the first de novo genome assembly of the surface fish morph using long-read sequencing technology. This approach yielded a much more comprehensive genome that allows for direct genome-wide comparisons between surface fish and Pachón cavefish. As a proof of principle, we first confirmed known genetic mutations associated with pigmentation and eye loss, then discovered novel quantitative trait loci (QTL), and identified coding and deletion mutations that highlight putative contributions to cave trait biology.

## Results

We set out to generate a robust reference genome for the surface morph of *A. mexicanus* from a single lab-reared female, descended from wild-caught individuals from known Mexico localities (Fig. 1a, Supplementary Fig. 1). We sequenced and assembled the genome using Pacific Biosciences single-molecule real-time (SMRT) sequencing (~73× genome coverage) and the wtdbg2 assembler[29] to an ungapped size of 1.29 Gb. Initial scaffolding of assembled contigs was accomplished with the aid of an *A. mexicanus* surface fish physical map (BioNano) followed by manual assignment of 70% of the assembly scaffolds to 25 total chromosomes using the existing *A. mexicanus* genetic linkage map markers[30]. The final genome assembly, Astyanax mexicanus

2.0, comprises a total of 2415 scaffolds (including single contig scaffolds) with N50 contig and scaffold lengths of 1.7 and 35 Mb, respectively, which is comparable to other similarly sequenced and assembled teleost fishes (Supplementary Table 1). The assembled regions (394 Mb) that we were unable to assign to chromosomes were mostly due to 3.36% (2235 markers total) of the genetic linkage markers not aligning to the surface fish genome, single markers per contig where the orientation could not be properly assigned, or markers that mapped to multiple places in the genome and thus could not be uniquely mapped. The uniquely mapped markers exhibited few ordering discrepancies and significant synteny between the linkage map[30] and the assembled scaffolds of the Astyanax mexicanus 2.0 genome, validating the order and orientation of a majority of the assembly (Fig. 1b; Supplementary Fig. 2). In Astyanax mexicanus 2.0, we assemble and identify 11% more total masked repeats than in the Astyanax mexicanus 1.0.2 assembly (Supplementary Table 1). Amongst a small sampling of assembled teleost genomes, *A. mexicanus* appears to be an intermediate for estimates of total interspersed repeats using WindowMasker[31] at 41%, compared to *Xiphophorus maculatus* (27%) and *Danio rerio* (50%). Possible mapping bias across cave population sequences to the cave vs surface fish genome references was also investigated by mapping population level resequencing reads to both genomes[32]. We found the number of unmapped reads is greater for all populations aligned to the Astyanax mexicanus 1.0.2 reference compared to Astyanax mexicanus 2.0 (Fig. 1c). Also, the percentage of properly paired reads, that is, pairs where both ends align to the same scaffold, is greater for all resequenced cave populations aligned to the Astyanax mexicanus 2.0 reference (Supplementary Fig. 3) and significantly more nonprimary alignments with greater variation were observed (Supplementary Fig. 4). Both metrics indicate that the Astyanax mexicanus 2.0 reference has more resolved sequence regions than the Astyanax mexicanus 1.0.2 reference. Future application of phased assembly approaches will likely resolve a significantly higher proportion of chromosomal sequences in the *A. mexicanus* genome[33].

Two independent sets of protein-coding genes were generated using the NCBI[34] and Ensembl[35] automated pipelines with similar numbers of genes found by each: 25,293 and 26,698, respectively (Supplementary Table 2). Gene annotation was aided by the diversity of transcript data derived from whole adult fish, embryos, and 12 different tissues available from the NCBI short-read archive. As a result, the total predicted protein-coding genes and transcripts (mRNA) were consistent with other annotated teleost species (Supplementary Table 2) and 1665 new protein-coding genes were added compared to Astyanax mexicanus 1.0.2. In addition, long noncoding gene (e.g., lncRNA) representation is significantly improved in the Astyanax mexicanus 2.0 reference compared to the Astyanax mexicanus 1.0.2 reference (5314 vs 1062), although targeted noncoding RNA sequencing will be required to achieve annotation comparable to zebrafish (Supplementary Table 2). We assessed completeness of gene annotation by applying benchmarking universal single-copy ortholog scores, which measure gene completeness among vertebrate genes. We find 94.6% of genes are complete, 4% are missing, and 4.2% are duplicated (Supplementary Table 3). In addition, using NCBIs transcript aligner Splign (C++ toolkit) gene annotation metrics, same-species RefSeq or GeneBank transcripts show 98.4% coverage when aligned to Astyanax mexicanus 2.0. In total, our measures of gene representation in the Astyanax mexicanus 2.0 reference show a high-quality resource for the study of *A. mexicanus* gene function.

Having a high-quality reference genome provides many benefits in exploiting *A. mexicanus* as a model species. Among the more important uses, from the standpoint of utilizing the system

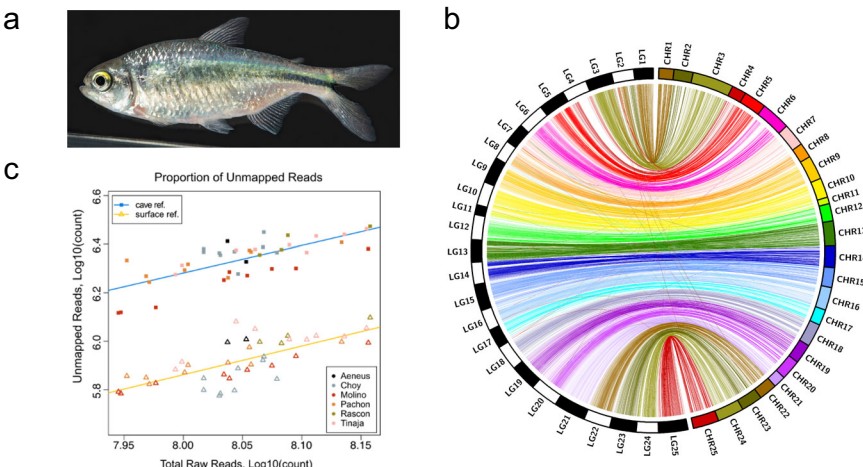

**Fig. 1 Assembly metrics for the surface fish genome Astyanax mexicanus 2.0. a** Adult *Astyanax mexicanus* surface fish. **b** Substantial synteny is evident between a recombination-based linkage map[30] and the draft surface fish genome. By mapping the relative positions of genotyping-by-sequencing (GBS) markers from a dense map constructed from a Pachón × surface fish $F_2$ pedigree, we observed significant synteny based on 96.6% of our genotype markers. **c** Proportion of unmapped reads for the same samples aligned to the cave (square points) and surface (triangle points) *Astyanax* reference genomes. Colors unite samples by population identity. Best fit lines for each alignment (cave: blue, surface: yellow) have similar slope, indicating that population identity has a similar effect on mapping rate to either genome.

to understand evolutionary mechanisms, are in mapping and identifying genes responsible for phenotypic change and in unveiling genomic structural variation (SV) that provides a substrate for adaptive selection. Thus, to demonstrate the advantages Astyanax mexicanus 2.0 brings to the field, we have explored its use in each of these settings, in particular examining the genetic underpinnings of albinism and reduced eye size, and then looking at the contribution of genomic deletion to variation within *A. mexicanus* populations.

First, with respect to identification of evolutionary important genes, we took advantage of a critical attribute of *A. mexicanus*, nearly unique among cave animals: the ability to interbreed cavefish and surface fish populations to generate fertile hybrids that can be intercrossed to perform QTL studies (for review see refs. [36,37]). The improved contig length of the surface fish genome and our syntenic analysis that unites the physical genome to a previously published linkage map based on genotyping-by-sequencing (GBS) markers[30] should greatly aid the correct identification of genetic changes linked to cave traits. To this end, we demonstrate the power of the Astyanax mexicanus 2.0 reference in gaining deeper insight into an already mapped trait (albinism) and in carrying out new QTL analysis of eye reduction.

Albinism has previously been mapped in QTL studies of both the Pachòn and Molino populations[20]. This work was carried out before the existence of any reference genome for the species. However, these mapping studies showed that a single major QTL in each population colocalized with a candidate gene *oca2*. *Oca2* loss-of-function mutations are known to cause albinism in other organisms, including humans, mice, and zebrafish, and deletions causing functional inactivation of Oca2 were identified in the albino fish from both the Molino and Pachón caves[20]. These compelling data indicating that *oca2* mutations are causal of albinism in cavefish were subsequently confirmed by CRISPR-mediated mutagenesis in surface fish[27]. To build on these results, we first performed two separate de novo QTL analyses of surface/Pachón $F_2$ hybrids (group A and B) in the context of the new reference genome (Supplementary Fig. 5a). Both studies identified a single QTL for albinism, with a LOD score of 47.31 at the peak marker on linkage group (LG) 3 (69% variance explained) in group A, and with LOD score of 22.56 at the peak marker on LG 21 (37.8% variance explained) in group B (Supplementary Fig. 5b, c, f, g). At

the peak QTL positions, $F_2$ hybrids homozygous for the cave allele are albino (Supplementary Fig. 5d, h). Mapping the markers associated with each of these LGs to the surface fish genome revealed that they are both located on surface fish chromosome 13 (Supplementary Table 4, Supplementary Fig. 5e, i). This demonstrates a significant improvement over mapping to the original cavefish genome: LG 3 from group A almost completely corresponds to surface fish chromosome 13 in Astyanax mexicanus 2.0, whereas it is split up into many contigs in Astyanax mexicanus 1.0.2 (Supplementary Figs. 5e, i and 6a). The QTL identified on LG 21 in group B corresponds to a 2.8 Mb region on surface fish chromosome 13 as determined using the sequences flanking the 1.5-LOD support interval as input for Ensembl basic local alignment search tool (Supplementary Table 4, Supplementary Fig. 5i). In line with previous mapping studies, the gene *oca2* is found within this region on surface fish chromosome 13. We further functionally verified that a change in the *oca2* locus is responsible for albinism in the $F_2$ hybrids, by crossing an albino surface/Pachón $F_2$ hybrid with a genetically engineered surface fish heterozygous for a deletion in *oca2* exon 21[27]. We found that 46.5% of the offspring completely lacked pigment ($n = 40/86$, Supplementary Fig. 5j). This confirms that a change in the *oca2* locus is responsible for loss of pigment in the mapping population.

While up until this point our reanalysis of albinism was largely confirmatory, an advantage of using the surface fish genome as a reference, compared to the previously available cavefish genome, is the ability to make comparisons between regions that are deleted in cavefish and as such are not available for sequence alignment. We utilized the Astyanax mexicanus 2.0 reference genome to align sequencing data obtained from wild-caught fish from different surface and cave localities[32] and analyzed the *oca2* locus (Supplementary Figs. 7 and 8). We found that nine out of ten Pachón cavefish carry the deletion in exon 24 that was previously reported in laboratory-raised fish (Supplementary Fig. 7[20]). None of the individuals from other populations for which sequence information was available carried the same deletion. Consistent with the laboratory strains, exon 21 of *oca2* is absent in all Molino cavefish samples ($n = 9$, Supplementary Fig. 8). None of the other sequenced population samples harbor the same deletion of exon 21; however, we found smaller heterozygous or homozygous deletions in exon 21 in some wild samples of Pachón ($n = 5/9$),

Río Choy ($n = 1/9$), and Tinaja (5/9) fish (Supplementary Fig. 8). In summary, we were able to detect deletions in *oca2* that would have not been discovered with alignments to Astyanax mexicanus 1.0.2 since exon 24 is missing in Pachón and thus no reference-based alignments were produced in that region.

We uncovered additional information by remapping previously published QTL[18–24] using the Astyanax mexicanus 2.0 reference. A total of 1060 out of 1124 markers (94.3%) mapped successfully with BLAST and were included in our surface fish QTL database. There were 77 markers that did not map to the cavefish genome[28] but did map to the surface fish genome, 52 markers that mapped to the cavefish but not the Astyanax mexicanus 2.0 reference, and 12 markers that did not map to either reference. The improved contiguity of the chromosome-level surface fish assembly allowed us to identify several additional candidate genes associated with QTL markers that were not previously identified in the more fragmented cavefish genome (Supplementary Data 1). For example, the markers Am205D and Am208E mapped to an ~1 Mb region of chromosome 6 of the surface fish genome (46,516,926–47,425,713 bp) but did not map to the Astyanax mexicanus 1.0.2 reference. This region is associated with feeding angle[21], eye size, vibration attraction behavior (VAB), suborbital neuromasts[23], and maxillary tooth number[19]. Multiple candidate genes related to cave-specific phenotypes are contained in this region including *rhodopsin* (vision), *ubiad1* (eye development), as well as *GABA A* receptor delta, which is associated with a variety of behaviors and could conceivably be involved in VAB. Notably, the scaffold containing these four genes in Astyanax mexicanus 1.0.2 (KB871939.1) was not linked to this QTL, demonstrating the utility of increased contiguity of Astyanax mexicanus 2.0.

Previous studies suggested that the gene *retinal homeobox gene 3* (*rx3*) lies within the QTL for outer plexiform layer of the eye[24]. Another QTL for eye size, size of the third suborbital bone, and body condition[18,19] may also contain *rx3* (low marker density and low power of older studies result in a broad QTL critical region in this area). The increased contiguity of Astyanax mexicanus 2.0 revealed that *rx3* is within the region encompassed by this QTL, whereas in Astyanax mexicanus 1.0.2, the marker for this QTL (Am55A) and *rx3* were located on separate scaffolds; thus, we could not appreciate that this QTL and key gene for eye development were in relatively close genomic proximity. While no amino acid coding changes are apparent between cavefish and surface fish, expression of *rx3* is reduced in Pachón cavefish relative to surface fish[28,38]. In zebrafish, *rx3* is expressed in the eye field of the anterior neural plate during gastrulation and has an essential role for the fate specification between eye and telencephalon[39,40]. We have compared *rx3* expression at the end of gastrulation and confirmed Pachón embryos have reduced expression domain size (Fig. 2a, b). The expression area is significantly smaller in Pachón embryos compared to stage-matched surface fish embryos. The expression of *rx3* is restored in F1 hybrids between cavefish and surface fish, indicating a recessive inheritance in cavefish (Fig. 2a, b). To test for a putative role of *rx3* in eye development in *A. mexicanus*, we used CRISPR/Cas9 to mutate this gene in surface fish (Fig. 2c), and assessed injected, crispant fish for eye phenotypes. Wild-type surface fish have large eyes (Fig. 2d). In contrast, externally visible eyes are completely absent in adult CRISPant surface fish ($n = 5$, Fig. 2d). Eye phenotypes were also reduced during development. Six days post fertilization *rx3* crispant fish have smaller eyes than fish injected with *Cas9* mRNA alone or uninjected siblings (Supplementary Fig. 9). This is consistent with work from other species, in which mutations in *rx3* (fish) or *Rx* (mice) result in a complete lack of eyes[41–43]. Together these data suggest that the role of *rx3* in eye development is conserved in *A. mexicanus*. Further, they support the hypothesis that regulatory changes in this gene may contribute to eye loss in cavefish through specification of a smaller eye field, and subsequently, production of a smaller eye.

In addition to the compiled database of older QTL studies, a number of genomic intervals associated with previously described locomotor activity difference between cavefish and surface fish[25] were also rescreened using the surface-anchored locations of markers from the high-density linkage map[30]. Within this Pachón/surface QTL map, we confirmed the presence of 20 previously reported candidate genes, and identified 96 additional genes with relevant GO terms, including *rx3*, further demonstrating the power and utility of this genomic resource (Supplementary Table 5). The new candidates include additional opsins

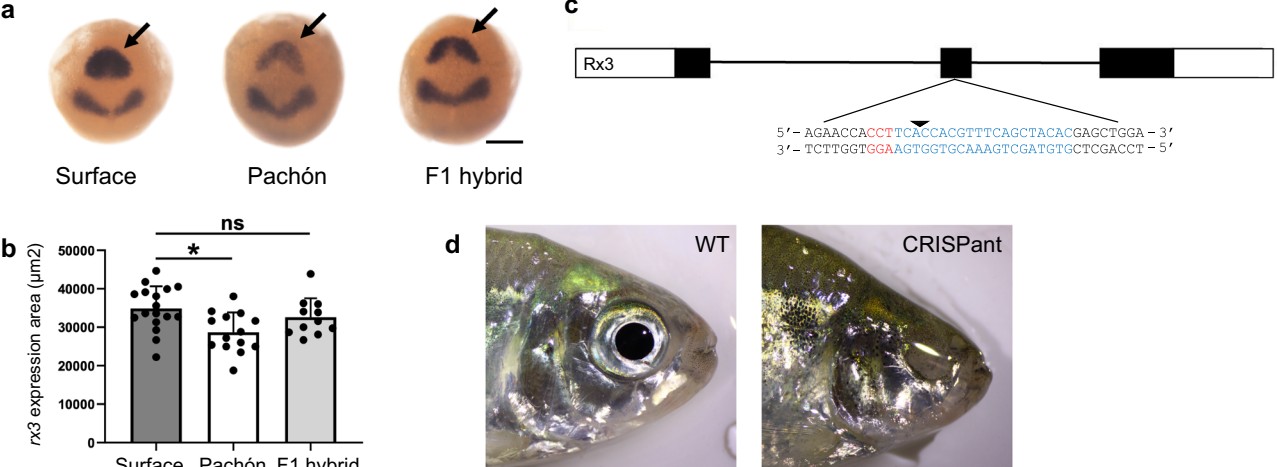

**Fig. 2 Rx3 analysis. a** Photographs of in situ hybridizations for *rx3* (arrows) and *pax2* mRNA at tailbud stage on surface fish, Pachón cavefish, and F1 hybrid between surface fish male and Pachón cavefish female. Scale bar: 200 μm. **b** Comparison of expression area of *rx3* at tailbud stage on surface fish (mean value 34876.3 μm², SD ± 5767.2, $n = 17$), Pachón (mean value 28674.8 μm², SD ± 5175.0, $n = 14$), and F1 hybrid (mean value 32617.8 μm², SD ± 4917.2, $n = 11$). Significance is shown for unpaired Student's *t* test (two-tailed); $p = 0.004$ is denoted by * and ns means not significant ($p = 0.294$). **c** Diagram of *rx3* gene. Boxes indicate exon and lines indicate introns. The empty boxes are UTR and the filled boxes are coding sequence (gene build from NCBI). A gRNA was designed targeting exon 2. The gRNA target site is in blue and the PAM site is in red. The arrow indicates the predicted Cas9 cut site. **d** Left: adult WT control (uninjected sibling) surface fish. Right: adult *rx3* CRISPR/Cas9-injected surface fish lacking eyes (CRISPant).

(*opn7a*, *opn8a*, *opn8b*, *tmtopsa*, and two putative green-sensitive opsins), as well as several genes contributing to circadian rhythmicity (*id2b*, *nfil3-5*, *cipcb*, *clocka*, and *npas2*). While analyses of expression data and sequence variation are necessary to determine which of these candidates exhibit meaningful differences between morphs, the presence of *clocka* and *npas2* in these intervals is of particular note, as the original analysis conducted using Astyanax mexicanus 1.0.2 did not provide any evidence of a potential role for members of the core circadian clockwork in mediating observed differences in locomotor activity patterns between Pachón and surface fish[15,25].

Finally, to gain new insight into the eye reduction phenotype, we used the Astyanax mexicanus 2.0 reference to de novo genetically map eye size in the two surface/Pachón F$_2$ groups that we used to map albinism. In surface/Pachón F$_2$ mapping population A ($n = 188$), we identified multiple QTLs for normalized eye perimeter that were spread across four LGs (Fig. 3a). The QTL on LG 1 is significant above a threshold of $p < 0.01$ (Fig. 3b). The surface allele at the peak marker appears to be dominant since eye size in the heterozygous state is similar to the

homozygous state (Fig. 3c). Mapping LG 1 to the Astyanax mexicanus 1.0.2 genome assembly results in markers under the QTL peak spread across different contigs, whereas these map to chromosome 3 in the surface fish assembly, emphasizing how the improved quality of the surface fish genome allows the identification of candidate genes throughout the QTL region (Fig. 3d, Supplementary Fig. 6b, Supplementary Table 4). This region has been identified previously[18,28] and contains genes such as *shisa2a* and *shisa2b*, as well as *eya1*. Analysis of the region between the markers with the highest LOD scores (3:9301997-9505868), however, revealed one gene that has not been linked to eye loss in *A. mexicanus* before, ENSAMXG00000005961 (Fig. 3d). Alignment of this novel gene sequence showed homology to orofacial cleft 1 (*ofcc1*), also called ojoplano (*opo*). In medaka, *opo* has been shown to be involved in eye development. When knocked out, *opo* affects the morphogenesis of several epithelial tissues, including impairment of optic cup folding which resulted in abnormal morphology of both the lens and neural retina in the embryos[44]. Sequence comparisons of this gene in Pachón cavefish and surface fish revealed several coding changes, however, none

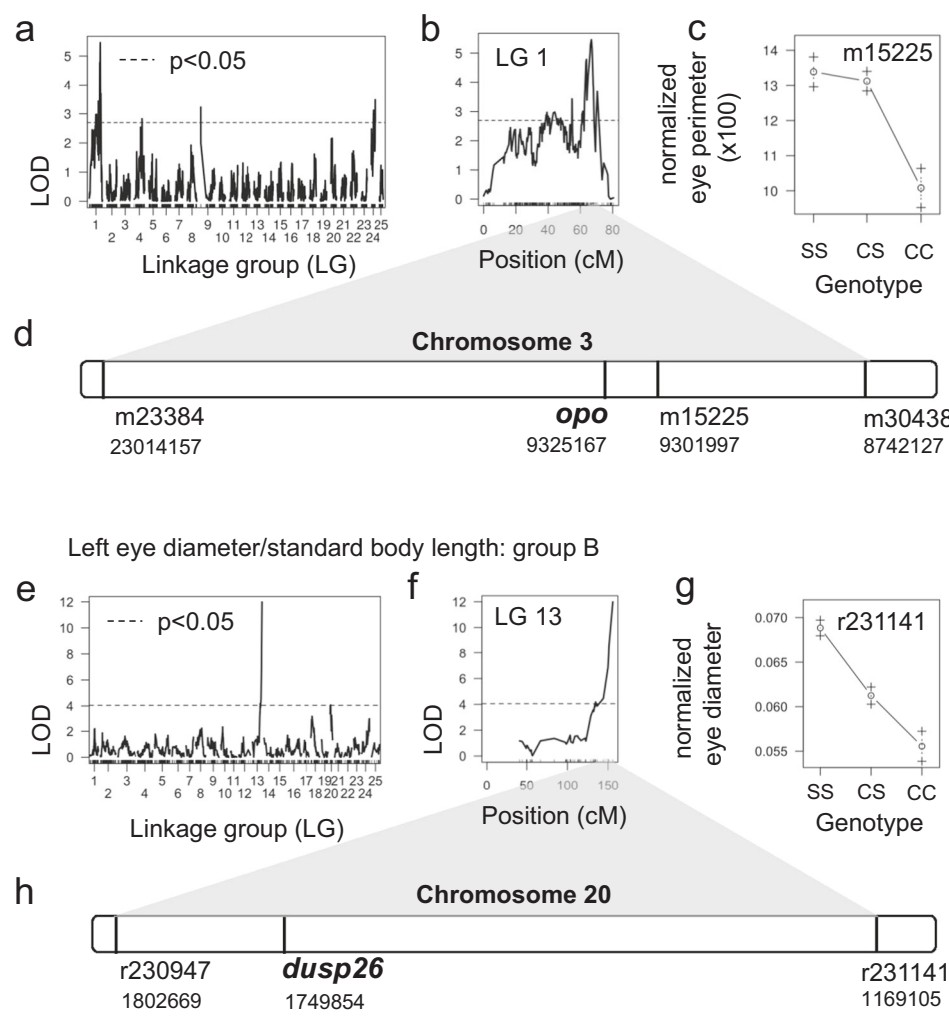

**Fig. 3 Utilizing the surface fish genome for QTL mapping of eye size in surface/Pachón hybrids.** Results from two independent mapping studies: group A (**a–d**) and group B (**e–h**). **a, e** Genome-wide LOD calculation for eye size using Haley–Knott regression and normal model. Significance threshold of 5% (black dotted line) determined by calculating the 95th percentile of genome-wide maximum penalized LOD score using 1000 random permutations. **b, f** LOD score for each marker on the linkage group with the peak marker. **c, g** Mean phenotype of the indicated genotype at the peak marker (S surface allele, C cave allele). **d** Relative position of the markers that define the 1.5-LOD support interval for group A on the surface fish chromosome 3. **h** Relative position of the markers that define the 1.5-LOD support interval for group B on chromosome 20 of the surface fish genome.

affect evolutionarily conserved residues. Future studies are needed to investigate a putative role of *opo* in the eye loss of Pachón cavefish.

In the surface/Pachón $F_2$ mapping population B ($n = 219$), we identified a single QTL for normalized left eye diameter on LG 13 with a LOD score of 11.98 at the peak marker that explains 24.3% of the variance in this trait (Fig. 3e, f). $F_2$ hybrids homozygous for the cave allele at this position have the smallest eye size, and the heterozygous state is intermediate (Fig. 3g). The values for left and right eye diameter mapped to the same region and, notably, we obtain the same peak when including eyeless fish in the map and coding eye phenotype as a binary trait (i.e., eyed, eyeless). The QTL on LG 13 corresponds to a 633-kb region on surface fish chromosome 13 as determined using the sequences flanking the 1.5-LOD support interval as input for Ensembl basic local alignment search tool (Fig. 3h, Supplementary Table 4). There are 22 genes in this region (Supplementary Table 6). Of note, none of the previously mapped eye size QTL in Pachón cavefish map to the same region[45]. A putative candidate gene in this interval is *dusp26*. Morpholino knockdown of *dusp26* in zebrafish results in small eyes with defective retina development and a less developed lens during embryogenesis[46]. We used whole-genome sequencing data[32] to compare the *dusp26* coding region between surface, Tinaja, Pachón, and Molino fish and found no coding changes. However, previously published embryonic transcriptome data indicate that expression of *dusp26* is reduced in Pachón cavefish at 36 hpf ($p < 0.05$) and 72 hpf ($p < 0.01$) (Supplementary Fig. 10)[47]. These data are in line with a potential role for *dusp26* in eye degeneration in cavefish; however, we cannot exclude critical contributions of other genes or genomic regions in the identified interval.

Another important advantage of having a robust reference genome is that it allows one to interrogate the structural variations (SV) at a population level. Knowledge of population-specific *A. mexicanus* structural sequence variation is lacking. Therefore, we aligned the population samples from Herman et al.[32] against Astyanax mexicanus 2.0 to ascertain the comparative state of deletions. We used the SV callers Manta[48] and LUMPY[49] to count the numbers of deletions present in each sample compared to Astyanax mexicanus 2.0 (Supplementary Fig. 11). While LUMPY tended to call a larger number of short deletions and Manta a smaller number of long deletions, there was high correlation ($R^2 = 0.78$) between the number of calls each made per sample (Supplementary Fig. 12), so we used the intersection of deletions called by both callers for further analysis. We then classified these deletions based on their effect (i.e., deletions of coding, intronic, regulatory, or intergenic sequence) (Supplementary Fig. 13).

Among the cavefish populations measured for deletion events, 412 genes contained deletions with an allele frequency > 5%. We found that the Molino population has the fewest heterozygous deletions, while Río Choy surface fish have the least homozygous deletions, mirroring the heterozygosity of single nucleotide polymorphisms[32] (Supplementary Fig. 13). In addition, the Tinaja population showed the most individual variability of either allelic state (standard deviation of 427). Pachón and Tinaja contained the highest number of protein-coding genes altered by a deletion in at least one haplotype, while Río Choy had the least (Fig. 4a). In two examples, *per3* and *ephx2*, we find the deletions that presumably altered protein-coding gene function varied in population representation, number of bases affected, and haplotype state for each (Fig. 4b, c). Of the 412 genes that contained deletions, 109 have assigned gene ontology in cavefish (Supplementary Table 7). We tested these 109 genes for canonical pathway enrichment using WebGestalt[50] and found genes significantly enriched ($p < 0.05$) for AMPK and MAPK signaling, as well as metabolic and circadian clock function (Table 1). In

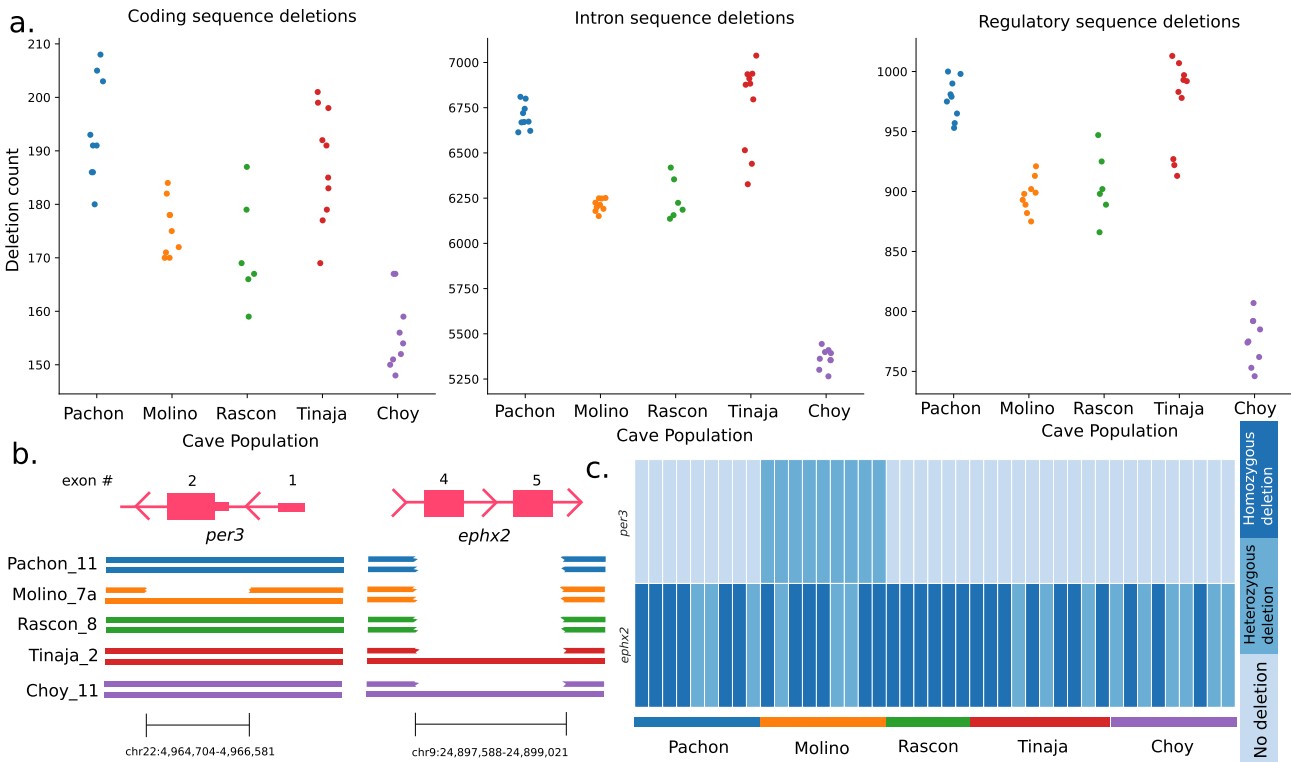

**Fig. 4 Sequence deletion events characterized by population origin. a** Total deletion count per population for protein-coding exons and intron sequence and regulatory sequence defined as 1000 bp up- or downstream of annotated protein-coding genes. **b** Gene sequence deletion coordinates for the circadian rhythm gene *per3* and cholesterol regulatory gene *ephx2* by population. **c** The allelic state of each deletion distributed by individual sample per population.

addition, some genes were linked to diseases consistent with cavefish phenotypes including: *ephx2*, which is linked to familial hypercholesterolemia, or *hnf4a*, which is linked with noninsulin dependent diabetes mellitus, based on the gene ontology of OMIM and DisGeNET[51]. Notably, these enrichment analyses recovered the deletion known to be the main cause of pigment degeneration in cavefish, *oca2*[27]. Further validation of these disease phenotype or pathway inferences is warranted.

We next used the surface fish genome to compare amino acid composition of key genes hypothesized to be involved in cave-specific adaptations. The wake-promoting hypothalamic neuropeptide hypocretin is a critical regulator of sleep in animals ranging from zebrafish to mammals[52–54]. We previously found that expression of hypocretin was elevated in Pachón cavefish compared to surface fish and pharmacological inhibitors of hypocretin signaling restore sleep to cavefish, suggesting enhanced hypocretin signaling underlies the evolution of sleep loss in cavefish[9]. In teleost fish, hypocretin signals through a single receptor, the hypocretin receptor 2 (Hcrtr2). We compared the sequence of *hcrtr2* in surface fish and Pachón cavefish and identified two missense mutations that result in protein-coding changes, S293V and E295K (Fig. 5a). To examine whether these were specific to Pachón cavefish or shared in other cavefish populations, we examined the *hcrtr2* coding sequence in Tinaja and Molino cavefish. The mutation affecting amino acid 295 is shared between Pachón and Tinaja populations (Fig. 5a, b). Further, we identified a 6 bp deletion that results in the loss of two amino acids (amino acid 140 and 141; SV) in Molino cavefish. The presence of these variants was validated by PCR and subsequent Sanger sequencing on DNA from individuals from laboratory populations of these fish. The E295K variant in Tinaja and Pachón as well as the two amino acid deletions in Molino affect evolutionarily conserved amino acids suggesting a potential impact on protein function (Fig. 5b). We performed an in silico analysis to test whether the identified cavefish variants in *hcrtr2* are predicted to affect the protein structure and stability of Hcrtr2. We used iStable[55] to test for potential destabilization effects of the substitution mutations found in Tinaja (E295K) and Pachón (S293V and E295K). iStable predicted a destabilization effect of E295K on the protein structure with a confidence score of 0.842. However, we found a stabilization effect of the S293V mutation in Pachón using iStable with a confidence score of 0.772. We repeated this analysis using MUpro[56] and obtained similar results. These findings raise the possibility that the evolved changes differentially affect the function of the same receptor. Structural changes in proteins upon amino acid deletion are difficult to predict with common tools such as iStable and MUpro. To analyze whether the deletion of S140 and V141 in the Molino *hcrtr2* could potentially influence the structural integrity of Hcrtr2, we performed a different analysis. We used the SWISS-MODEL protein structure prediction tool[57] to identify potential differences in the protein structure between surface and Molino Hcrtr2. We modeled the surface and Molino Hcrtr2 protein using crystal data from the human HCRTR2 (5wqc[58]). We then used the VMD visualization software to overlay the surface fish and Molino predicted structure. This analysis indicates that the deletion of S140 and V141 disrupts the structural integrity of an alpha helical structure in the transmembrane region of Hcrtr2 that could potentially affect the stability of this receptor (Fig. 5c). We also tested the Tinaja and Pachón *hcrtr2* sequences using a similar approach and found only minor differences between the respective cavefish Hcrtr2 structure when overlaid with the surface fish structure (Fig. 5c). To confirm these potential structural changes in cavefish Hcrtr2 advanced in situ and in vivo protein, analysis need to be performed in future studies; however, the identification of coding mutations in three different populations of cavefish supports the notion that hypocretin signaling is under selection at the level of receptor. This is in line with population sequencing data[32], which identified *hcrtr2* in the top 5% of FST outliers between surface fish (Rascon) and Pachón cavefish.

## Discussion

The genomic and phenotypic surface-to-cave transitions in *A. mexicanus* serve as an indispensable model for the study of natural polygenic trait adaptation. Here, we present a high-quality "chromonome" of the surface form of *A. mexicanus*. Our surface fish genome far surpasses the contiguity and completeness compared to the assembled cavefish version of *A. mexicanus*[28] owing to the use of updated long-read and mapping technology. This allowed for substantial increases in the detected sequence variation associated with cave phenotypes. A substantial level of heterozygosity within surface fish populations, coupled with our use of two surface populations for de novo assembly, however, prevented more complete sequence connectivity as compared to similarly assembled fish genomes such as *Xiphophorus maculatus*[59], a laboratory lineage of reared fish (Supplementary Table 1). Nevertheless, a 40-fold reduction in assembled contigs (an indirect measure of gaps), a 71-fold improvement in N50 contig length, high-level synteny with prior linkage map measures of *A. mexicanus* chromosome order, and the addition of 1665 protein-coding genes all offer researchers a resource for more complete genetic experimentation. In this study, we explored cavefish genomes of diverse origins relative to this new genomic proxy for the ancestral (surface) state. This yielded numerous discoveries when determining evolutionary-derived sequence features that may be driving cave phenotypes.

We have now refined our knowledge of the genetic basis of troglomorphic traits by identifying candidate genes within new and previously identified QTL. Trait-associated sequence markers, when placed on more contiguous chromosomes of the

**Table 1 Protein-coding genes altered by deletion events and their enrichment among canonical pathways and disease.**

| Gene pathway or disease association | Genes | Database | p value |
|---|---|---|---|
| AMPK signaling | *hnf4a;lipe;ppp2r5a* | KEGG | 0.01 |
| Wnt signaling | *csnk2b;hltf;map3k7cl;pcdh9;ppp2r5a* | Panther | 0.01 |
| MAPK signaling | *gadd45a;mapkapk3* | Panther | 0.01 |
| Metabolism | *cmpk1;cmpk2;ephx2;mtmr6;ndufa4l2;pigp;prps1;sgms1;sms;uxs1* | KEGG | 0.02 |
| **Circadian clock** | **per3** | **Panther** | **0.05** |
| **Hypercholesterolemia** | **ephx2** | **OMIM** | **0.005** |
| **Diabetes mellitus** | **nnf4a** | **OMIM** | **0.02** |
| **Albinism** | **pp3b1;oca2** | **DisGeNET** | **0.002** |
| **Sialorrhea** | **ikzf1;prps1;vps13a** | **DisGeNET** | **0.001** |

Bold text denotes significant tests for disease enrichment. For enrichment tests, we used a hypergeometric test, and the significance level was set at 0.05 and implemented the Benjamini and Hochberg multiple test adjustment[88] to control for false discovery.

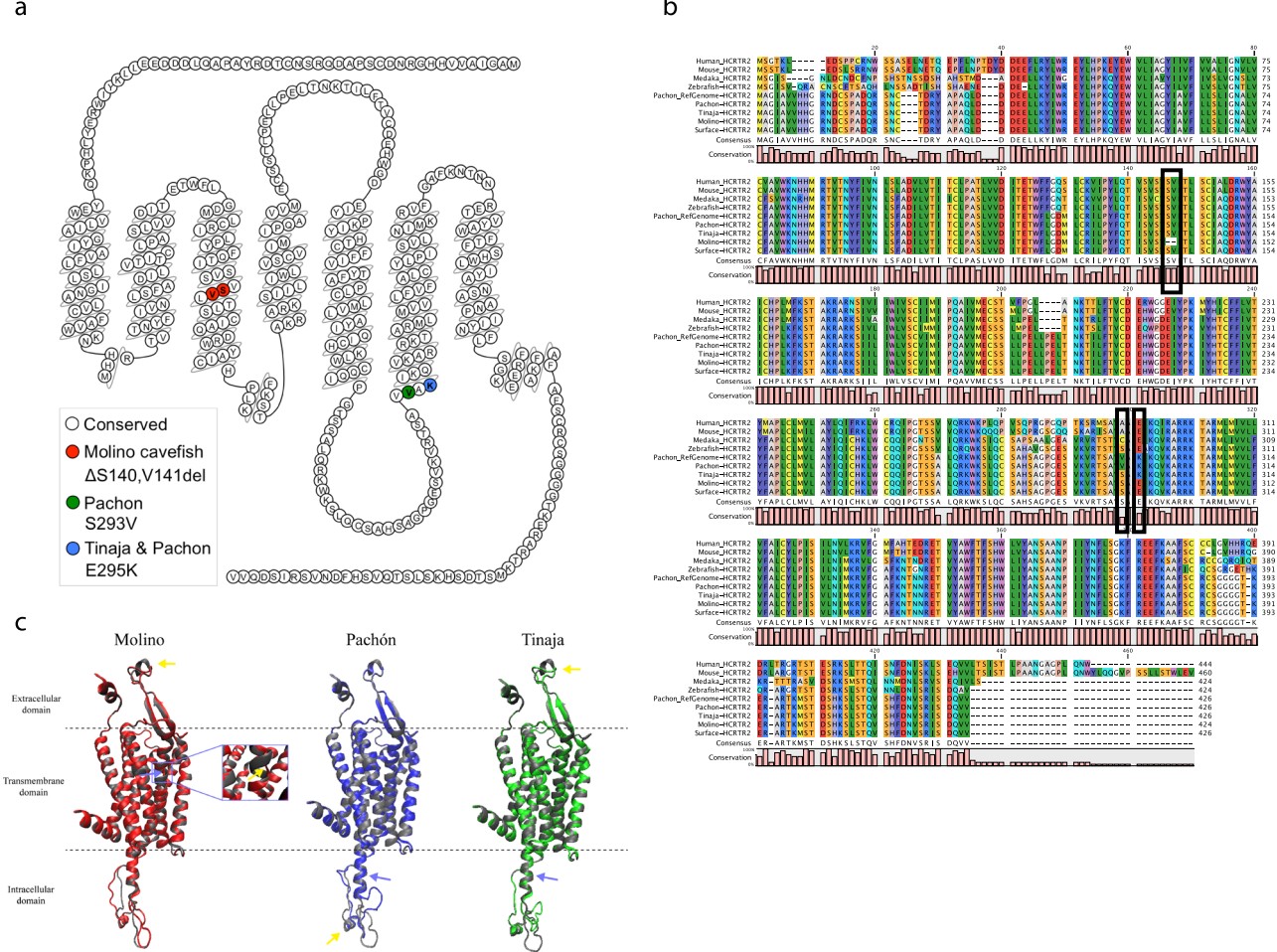

**Fig. 5 Hcrtr2 mutations in cavefish. a** Position of mutations in *hcrtr2* in the different cavefish populations. **b** Sequence alignment showing that the mutations in Molino and Tinaja are affecting evolutionary conserved amino acids. **c** 3D model based on structure of human HCRTR2. Structure is displayed as comic structure displaying alpha helices and beta sheets using VMD 1.9.3. A model of Hcrtr2 from surface and cavefish fish was rendered using SWISS-MODEL (https://swissmodel.expasy.org/) and X-ray crystal data from RCSB Protein Data Bank: 5wqc. Purple arrows indicate site of mutation and yellow arrows indicate sites with differences in the secondary structure between surface and respective cavefish. In the Molino projection, the deletion (S140 and V141) in the Molino sequence causes a break in one of the alpha helix structures of the transmembrane domain (highlighted in purple box). These images were made with VMD[89] software support. VMD is developed with NIH support by the Theoretical and Computational Biophysics group at the Beckman Institute, University of Illinois at Urbana-Champaign.

surface fish genome, delineate sequence structure, especially gene regulatory elements. As proof of this added discovery potential, we identified new candidate genes associated with eye loss relying on a single-QTL model. Previous studies using multi-QTL models have identified eight QTL associated with eye size in Pachón showing that eye degeneration likely involves a number of genes[19]. Part of the single eye size QTL we mapped in one of our Pachón/surface populations had been previously identified, but our analysis shifted the LOD peak to just outside the previously identified region, revealing a new candidate gene for eye loss, *opo*. Genetic mapping in a second surface/Pachón hybrid population revealed a new QTL for eye size. A gene within this region, *dusp26*, has lower expression in Pachón cavefish during critical times of eye development. New studies aimed at identifying regulatory changes linked to these candidates will further clarify the role of these genes in the evolution of eye size reduction.

SV such as genomic deletions represent a significant source of standing variation for trait adaptation[60], which we were unable to accurately measure previously due to numerous assembly gaps in the *Astyanax mexicanus* 1.0.2 genome assembly. A species' use of SV, including deletions, insertions, inversions, and other more complex events, can enable trait evolution, but their occurrence and importance in phenotypic adaptation among wild populations is understudied. A small number of studies in teleost fish show the standing pool of SVs outnumbers single-nucleotide variations and are likely in use to alter the genotype to phenotype continuum of natural selection[61,62]. For example, rampant copy number variation occurs during population differentiation in stickleback[60]. We conducted the first study of cavefish SVs, albeit only identifying deletions. We identified SVs possibly contributing to observed troglomorphic phenotypes, discovered previously unidentified population-unique deletions, and broadly classified their putative impact through gene function inference to zebrafish. Of the deletion altered protein-coding genes where biological inference is possible, we find low-and high-frequency deletions in *per3* and *ephx2*, respectively (Fig. 5b, c). Some of these gene disruptions fit expected cave phenotypes, while others are of unknown significance (e.g., *sms*, a gene involved in beta-alanine metabolism). *per3* plays a role in vertebrate circadian regulation and corticogenesis[63], and sleep and circadian patterns are significantly disrupted in cavefish[9]. In all sampled Molino fish, *per3* harbors a heterozygous deletion (Fig. 4b, c).

In addition, we identify multiple evolved coding changes within *hcrtr2*, a receptor that regulates sleep across vertebrate species[64]. This study uncovered unique changes in cave populations at the *hcrtr2* locus. Future genome-wide analysis of G protein-coupled receptor (GPCR) differences between surface fish and cavefish populations, combined with functional studies using allele swapping between cavefish and surface fish, will further elucidate the relationship between GPCR function and trait evolution.

Our discovery of novel cave-specific features of several genes and many others not explored here highlight novel insights that were obscured when using Astyanax mexicanus 1.0.2 genome assembly as a reference. The first cave population-wide catalog of thousands of deletions that could potentially impact the cave phenotypes collectively awaits investigation and validation. Future assembly improvements to both the cave and surface forms of the *A. mexicanus* genome are expected and will be key in understanding population-genetic processes that enabled cave colonization. The shared gene synteny of *A. mexicanus* with zebrafish, gene editing success demonstrated in this study, and their interbreeding capabilities promise to reveal not only genic contributions, but likely regulatory regions associated with the molecular origins of troglomorphic trait differences.

## Methods

**Ethics oversight.** The experiments were approved by the institutional animal care and use committees of the institutes that were part of this study. The following animal protocols cover the experiments described in this study: Stowers Institute for Medical Research: 2019-084; Florida Atlantic University: A20-11, A17-35; Harvard Medical School: IS00001612-3; and University of Oxford: PPL 30/3258.

### Genome sequencing and assembly curation

*DNA sequencing.* High molecular weight DNA was isolated from a single female surface fish using the MagAttract kit (Qiagen) according to the manufacturer's protocol. In order to obtain enough high-quality DNA and to sample broader genetic variation, we chose a large female fish (Asty152) from a lab-reared cross of wild-caught *A. mexicanus* surface populations from the Río Sabinas and the Río Valles surface localities (Supplementary Fig. 1a). SMRT sequencing was completed on a PacBio RSII instrument, yielding an average read length of ~12 kb. SMRT sequence coverage of >50-fold was generated based on an estimated genome size of 1.3 Gb. All SMRT sequences are available under NCBI BioProject number PRJNA533584.

*Assembly and error correction.* For de novo assembly of all SMRT sequences, we used a fuzzy de Bruijn graph algorithm, wtdbg[29], for contig graph construction, followed by collective raw read alignment for assembly base error correction. All error-prone reads were first used to generate an assembly graph with genomic k-mers unique to each read that results in primary contigs. Assembled primary contigs were corrected for random base error, predominately indels, using all raw reads mapped with minimap2[65]. To further reduce consensus assembly base error, we corrected homozygous insertion, deletion, and single base differences using default parameter settings in Pilon[66] with ~60× coverage of an Illumina PCR-free library (150 bp read length) derived from Asty152 DNA.

*Assembly scaffolding.* To scaffold de novo assembled contigs, we generated a Bio-Nano Irys restriction map of another surface fish (Asty168) that allowed sequence contigs to be ordered and oriented, and potential misassemblies to be identified. Asty168 is the offspring of two surface fish, one from the Río Valles (Asty02) and the other from the Río Sabinas locality (Asty04), both Asty02 and Asty04 were the offspring of wild-caught fish (Supplementary Fig. 1b). We prepared HMW-DNA in agar plugs using a previously established protocol for soft tissues[67]. Briefly, we followed a series of enzymatic reactions that (1) lysed cells, (2) degraded protein and RNA, and (3) added fluorescent labels to nicked sites using the IrysPrep Reagent Kit. The nicked DNA fragments were labeled with Alexa Fluor 546 dye, and the DNA molecules were counter-stained with YOYO-1 dye. The labeled DNA fragments were electrophoretically elongated and sized on a single IrysChip, and subsequent imaging and data processing determined the size of each DNA fragment. Finally, a BioNano proprietary algorithm performed a de novo assembly of all labeled fragments > 150 kbp into a whole-genome optical map with defined overlap patterns. The individual map was clustered and scored for pairwise similarity, and Euclidian distance matrices were built. Manual refinements were then performed as previously described[67].

*Chromosome builds.* Upon chimeric contig correction and completion of scaffold assembly steps using the BioNano map, we used Chromonomer[68] to align all possible scaffolds to the *A. mexicanus* high-density linkage map[30], then assigned chromosome coordinates. Using default parameter settings, Chromonomer attempts to find the best set of nonconflicting markers that maximizes the number of scaffolds in the map, while minimizing ordering discrepancies. The output is a FASTA file format describing the location of scaffolds by chromosome: a "chromonome".

*Defining syntenic regions between cave and surface genomes.* A total of 2235 GBS markers[30] were mapped to both the cave (Astyanax mexicanus 1.0.2) and surface fish (Astyanax mexicanus 2.0) assemblies (Supplementary Fig. 2). These GBS markers were mapped to Astyanax mexicanus 1.0.2 using the Ensembl "BLAST/BLAT search" web tool and resulting information from each individual query was transcribed into an Excel worksheet. To map to the surface fish genome, the NCBI "Magic-BLAST" (version1.3.0) command line mapping tool[69] was used. The resulting output of the mapping was a single, tabular formatted spreadsheet, which was used to visualize syntenic regions from the constructed linkage map[30]. We used Circos software to visualize the positions of all markers that mapped to the Astyanax mexicanus 2.0 genome[70]. Any chromonome errors detected through these synteny alignments were investigated and manually corrected if orthologous data were available, such as Astyanax mexicanus 1.0.2 scaffold alignment. Moreover, to facilitate future investigations into the location of previously discovered QTL in each form, we provide the corresponding coordinates of the assembly to assembly alignments for Astyanax mexicanus 1.0.2 and Astyanax mexicanus 2.0 (Supplementary Data 1).

*Gene annotation.* The Astyanax mexicanus 2.0 assembly was annotated using the previously described NCBI[71] and Ensembl[35] pipelines, including masking of repeats prior to ab initio gene predictions and RNAseq evidence-supported gene model building. NCBI and Ensembl gene annotation relied on an extensive variety of publicly RNAseq data from both cave and surface fish tissues to improve gene model accuracy. The Astyanax mexicanus 2.0 RefSeq or Ensembl release 98 gene annotation reports each provide a full accounting of all methodology deployed and their output metrics within each respective browser.

*Assaying genome quality using population genomic samples.* To understand the impact reference sequence bias and quality may have on downstream population genomic analyses for the cavefish and surface fish genomes, we utilized the population genomic resequenced individuals processed in Herman et al.[32]. In brief, 100 bp sequences were aligned to the reference genomes Astyanax mexicanus 1.0.2 and Astyanax mexicanus 2.0. The NCBI annotation pipeline of both assemblies included WindowMasker and RepeatMasker steps to delineate and exclude repetitive regions from gene model annotation. The positional coordinates for repeats identified by RepeatMasker are provided in the BED format at NCBI for each genome. WindowMasker's "nmer" files (counts) were used to regenerate repetitive region BED coordinates[31]. The BED coordinates for both maskers were intersected with BEDTools v2.27.1[72]. We used SAMtools 1.9 with these coordinates and alignment quality scores to filter alignments and generate summary statistics for each sample aligned to the cave and surface reference genomes[73]. Summary plots were generated in R 3.6.

*Genetic mapping in surface × Pachón crosses.* To map albinism and eye size to the new surface genome and test robustness in this methodology across laboratories, two independent $F_2$ mapping populations consisting of surface/Pachón hybrids were analyzed. First, we scored 188 surface/Pachón $F_2$ hybrids for albinism and normalized eye perimeter that were used in a previously published genetic mapping study[10]. Phenotypes were assessed using macroscopic images of entire fish and measurements were obtained using ImageJ[74]. Normalized eye perimeter was determined by dividing eye perimeter by body length. Albinism was scored as absence of body and eye pigment.

The 25 LGs constructed de novo in earlier studies of this population[10] were scanned using the R (v.3.5.3) package R/qtl (v.1.44-9)[75] using the scanone function for markers linked with albinism (binary model) or left eye perimeter relative to body length (normal model). The genome-wide LOD significance threshold was set at the 95th percentile of 1000 permutations. All marker sequences were aligned to both the Astyanax mexicanus 1.0.2 and Astyanax mexicanus 2.0 references using Bowtie (v.2.2.6 in sensitive mode)[76]. Circos plots were generated using v.0.69-6[70].

The second $F_2$ mapping population ($n = 219$) consisted of three clutches produced from breeding paired $F_1$ surface/Pachón hybrid siblings[77]. Albinism and eye size were assessed using macroscopic images of entire fish. Eye diameter and fish length were measured in ImageJ[74] according to ref. [78]. Fish lacking an eye (21/195 on the left side, 22/172 on the right side) were not included in the analysis of eye size in order to analyze eye size using a normal distribution model. We found that fish length was positively correlated with eye diameter. To eliminate the effect of fish length on potential QTL, we analyzed eye diameter normalized to standard length.

We used R/qtl[75] to scan the LGs (scanone function) for markers linked with albinism (binary model) or eye diameter relative to body length (normal model) and assessed statistical significance of the LOD scores by calculating the 95th percentile of genome-wide maximum penalized LOD score using 1000 random

permutations. We estimated confidence intervals for the QTL using 1.5-LOD support interval (Iodint function).

*Complementation analysis in albino surface–Pachón F₂ fish.* An albino surface–Pachón F₂ fish was crossed to $oca2^{\Delta4bp/+}$ surface fish[27]. Five day post fertilization, larval offspring from this cross were scored for pigmentation (presence or absence of melanin pigmentation) by routine observation. Larvae were imaged under a dissecting microscope and the number of pigmented and albino progeny were assessed. Following this, DNA was extracted from eight pigmented and eight albino progeny, and these fish were then genotyped for the engineered 4 bp deletion by PCR followed by gel electrophoresis, using locus specific primers (Supplementary Table 8—Complementation Primer) and methods[79,80]. Briefly, larval fish were euthanized in MS-222, fish were imaged, then DNA was extracted from whole fish, PCR was performed followed by gel electrophoresis. All genotyped albino embryos had two bands (indicating the presence of the engineered deletion), whereas all genotyped pigmented embryos had a single band, indicating inheritance of the wild-type allele from the surface fish parent.

*Genetic mapping of previously mapped QTL studies to the surface fish genome.* To identify any candidate genes potentially associated with cave phenotypes that were not evident in the more fragmented Astyanax mexicanus 1.0.2 genome, we generated a QTL database for the Astyanax mexicanus 2.0 genome to identify genomic regions containing groups of markers associated with cave-derived phenotypes. We followed the methods[32] used to create a similar database for the cavefish genome. BLAST was used to identify locations in the surface fish genome for 1156 markers from several previous QTL studies[18,19,21–23,81]. The top BLAST hit for each marker was identified by ranking e-value, followed by bitscore, and then alignment length. Previously, 687 of these markers were mapped to the cavefish genome. BEDTools intersect and the surface fish genome annotation was used to identify all genes within the regions of interest. Genomic intervals associated with activity QTL previously identified using the existing high-density linkage map were similarly reexamined using the established locations of the 2235 GBS markers within the Astyanax mexicanus 2.0 genome[25]. We then investigated whether genes identified near QTL had ontologies associated with cave phenotypes using the NCBI (https://www.ncbi.nlm.nih.gov/) or Ensembl genome browser (https://www.ensembl.org). However, genes were referred to as candidates because they are within the QTL confidence interval. Even if a gene within the interval does not have previously known functions contributing to the trait of interest, it does not eliminate it as a potential candidate.

*Generation of rx3 CRISPant fish.* CRISPant surface fish were generated using CRISPR/Cas9. A gRNA targeting exon 2 was designed and generated as described previously[82]. Briefly, oligo A containing the gRNA sequence (5′-GTGTAGCT-GAAACGTGGTGA-3′) between the sequence for the T7 promoter and a sequence overlapping with the second oligo, oligo B, was synthesized (IDT) (Supplementary Table 8—CRISPR-Oligo). Following annealing with Oligo B and amplification, the T7 Megascript Kit (Ambion) was used to transcribe the gRNA with several modifications, as in refs. [27,80]. Specifically, the manufacturer's directions were followed except that the following reagents were added per transcription reaction: 10 μl of RNAse-free water, 5 μl of DNA template, 1 μl of each nucleoside triphosphate, 1 μl of 10x transcription buffer, and 1 μl of T7 polymerase enzyme mix. The gRNA was cleaned up using the miRNeasy mini kit (Qiagen) following manufacturer's instructions and eluted into RNase-free water. Nls-Cas9-nls[83] mRNA was transcribed using the mMessage mMACHINE T3 kit (Life Technologies) following manufacturer's directions. Single-cell embryos were injected with 25 pg of gRNA and 150 pg Cas9 mRNA. Injections were performed using a capillary glass needle mounted in a micromanipulator connected to a microinjector. Injection pressure was adjusted with every needle to achieve a ~1.0 nl injection volume[79,80]. Injected fish were screened to assess for mutagenesis by PCR using primers surrounding the gRNA target site (Supplementary Table 8—CRISPant primer). Genotyping was performed with DNA extracted from whole larval fish from the clutch, as described previously[84]. Briefly, DNA was isolated by incubating samples in 50 mM NaOH at 95 °C for 30 min, after which a 1/10th volume of Tris-HCL pH 8 was added. Following PCR, gel electrophoresis was used to discriminate between alleles with indels (more than one band results in a smeary band) and wild-type alleles (single band: as in ref. [79]). For adult fish, injected fish (crispants) along with uninjected wild-type siblings were raised to adulthood. Eye size was assessed in both rx3 gRNA/Cas9 mRNA injected fish (crispants) and wild-type sibling adult fish (n = 5 each). Fish were anesthetized in MS-222 and imaged under a dissecting microscope. For larval fish, eye size was assessed at day 6 post fertilization and compared between rx3 CRISPant fish, 150 pg Cas9 mRNA only injected siblings, and wild-type uninjected siblings (n = 10 each) using Fiji[74] and corrected for standard length. Eye area was compared between groups using an ANOVA followed by post hoc Tukey tests. Statistics were performed in GraphPad Prism.

*Whole mount in situ hybridization and imaging.* Full length of rx3 coding sequence was amplified by PCR from surface fish cDNA (Supplementary Table 8—WMISH-primer). The antisense probe was generated using a digoxigenin RNA Labeling Kit (Roche Diagnostics, Indianapolis, IN). In situ hybridization was performed with 65% formamide of the hybridization mix, and the incubation temperature was set

at 65 °C. The stained embryos were dehydrated through a successive series of methanol and stored at −20 °C. Before imaging, the embryos were gradually rehydrated into PBST (PBS with 0.1% Tween-20). During the imaging, the embryos were suspended in 0.2% agar in PBS, and the expression area of rx3 was measured using NIS Elements F 2.30 software with a stereomicroscope (SMZ1500, Nikon).

*SV detection.* We first aligned Illumina reads from *A. mexicanus* population resequencing data described by Herman et al.[32] to the Astyanax mexicanus 2.0 (SRA accession: PRJNA260715) reference using BWA-MEM v0.7.17[85] with default options. We then converted the output to bam format, fixed mate pair information, sorted and removed duplicates using the SAMtools v1.9 view -bh, fixmate -m, sort, and markdup -r modules, respectively[73]. We ran two software packages for calling structural variants using these alignments: manta[48] and lumpy[49]. We ran manta v1.6.0 with default options as suggested in the package's manual, and lumpy using the smoove v0.2.3 pipeline (https://github.com/brentp/smoove) using the commands given in the "Population calling" section. Nextflow[86] workflows for our running of both SV callers can be found at https://github.com/esrice/workflows. Due to the low sequence coverage (average ~9×) available per sample, we confined analysis to deletions called by both SV callers and within the size range 500–100 kb. We consider a deletion to be present in both sets of calls if there was a reciprocal overlap of 50% of the length of the deletion. We used the scripts merge_deletions.py to find the intersection of the two sets of deletions, annotate_vcf.py to group the deletions by their effect (i.e., deletion of coding, intronic, regulatory, or intergenic sequence), and count_variants_per_sample.py to count the numbers of variants called in each sample; all scripts are publicly available (see Code Availability). All protein-coding genes with detected deletions among cavefish populations as defined in this study were used as input to test for significant enrichment among specific databases within WebGestalt[50]. Entrez gene IDs were input as gene symbols, with organism of interest set to zebrafish using protein-coding genes as the reference set. Pathway common databases of KEGG and Panther canonical gene signaling pathways as well as the various genes associated with diseases curated in OMIM and Digenet[87] were reported using a hypergeometric test, and the significance level was set at 0.05. We implemented the Benjamini and Hochberg multiple test adjustment[88] to control for false discovery.

**Reporting summary.** Further information on research design is available in the Nature Research Reporting Summary linked to this article.

## Data availability
The raw sequencing and final assembly data generated in this study have been deposited in the NCBI BioProject database under accession code PRJNA89115. The resequencing data used in this study are available in the NCBI BioProject database under accession code PRJNA260715. Original data underlying this manuscript can be accessed from the Stowers Original Data Repository at http://www.stowers.org/research/publications/libpb-1528, found within the article or made available by the authors on request.

## Code availability
All codes used to generate the data in this paper are available on GitHub: https://doi.org/10.5281/zenodo.4433170.

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

## Acknowledgements

The authors would like to thank Karin Zueckert-Gaudenz and Mihaela Sardiu for technical assistance and the Stowers aquatics group for fish care and help with shipments of fish between NYU and Stowers. A.K., S.E.M., and N.R. are supported by NIH 1R01GM127872-01. This work was also supported by NIH R24OD011198 to W.C.W., L.H., E.S.R., T.G.-L., and M.K., NSF EDGE Award 1923372 to N.R., J.E.K., and S.E.M., NSF DEB-1754231 to J.E.K. and A.K., and NSF IOS-1933428 to J.E.K., S.E.M., and N.R. Author N.R. is further supported by institutional funding, funding from the Edward Mallinckrodt Foundation, and NIH DP2AG071466. J.E.K. is further supported by NIH R15HD099022. J.B.G. is supported by NIDCR R01-DE025033 and NSF DEB-1457630. Some computation for this work was performed on the high-performance computing infrastructure provided by Research Computing Support Services and in part by the National Science Foundation under grant number CNS-1429294 at the University of Missouri, Columbia, MO, USA. The Minnesota Supercomputing Institute at the University of Minnesota provided resources that contributed to the research results reported within this paper.

## Author contributions

W.C.W. and N.R. conceived of the study. All authors performed and analyzed the experiments. W.C.W., N.R., R.B., B.M.C., J.B.G., A.C.K., J.E.K., S.E.M., M.T.M.M., R.P., M.R.R., C.T., and Y.Y. wrote the paper.

## Competing interests

The authors declare no competing interests.
