## [Peer Review File · Nature Communications]

REVIEWER COMMENTS

Reviewer #1 (Remarks to the Author):

The manuscript "A chromosome level genome of *Astyanax mexicanus* surface fish for comparing population specific genetic differences contributing to trait evolution." By Warren et al describes a substantially improved assembly, which is used to resolve several biologically interesting phenomena in the species. This is an excellent piece of work, and I only have a couple of comments related to the clarity and consistency of presentation of the genome and QTL mapping data.

1) The scheme used for naming the chromosomal scaffolds. It is not clear what criteria were used to name the scaffolds. Under standard nomenclature chromosome 1 would correspond to the longest chromosome in the karyotype, chromosome 2 the second longest and so forth. However, I did not see where the assembly was anchored to physically defined chromosome although it is clearly anchored to linkage groups. Given the data the authors appear to have on hand the scaffolds might be more appropriately named LG1, LG2 ... if they are ordered based on linkage distance or scaff1, scaff2 ... if they are ordered based on size.

2) Unifying the linkage analyses with this nomenclature will also make the presentation of the results more straight-forward. In some cases this seems to have been done, but in others (e.g. in Figure 2 where the same genomic interval appears to be referred to as LG3, LG21 and chr13, and similarly in Figure 4). It would seem that these analyses would be clearer if they were all performed/presented using the same underlying linear scaffold (chr13, or an appropriately named version of this same scaffold).

3) A supplemental table showing the correspondence between their chromosomal genomic scaffolds, previous (relevant) linkage groups and other previous or parallel (e.g. NCBI accession numbers) assemblies might also be useful given the history of this system

Sincerely,
Jeremiah Smith

Reviewer #3 (Remarks to the Author):

This manuscript reported the first chromosome-level genome assembly of the emerging evolutionary and medical model of the Mexican tetra, *Astyanax mexicanus*. Authors build this genome assembly based on the surface population of *A. mexicanus*, and re-sequenced 4 of cave populations. By utilizing it, authors addressed deletion or loss of the genes in the derived cave populations comparing with ancestral-type surface population and zebrafish. From the view of the full genome, they retested the albinism gene (*Oca2*), and newly tested *Rx3* for eye degeneration. Also, the genetic and genomic bases on cavefish traits (other candidate genes for eye degeneration, insomnia, metabolism, and circadian clock) were surveyed.

Many of these results were novel and a few were reanalyses (*Oca2*, Protas 2005; candidate eye genes, McGough 2013). Their results advance the knowledge of the complex evolutionary processes for cave adaptation. These results will impact cave ecology and biology and in evolutionary biology of rapid adaptation.

Their logic was clear and presented in a proper way.

Because cavefish is an established laboratory model which has ancestral-morph, and polarity of evolution (from surface-type ancestor to cave-morph) and major selection pressures were well defined, the genetics bases for morphological, metabolic and behavioral changes provided in this paper are, I expect, influential to the cave biology and evolutionary biology fields.

There is one major concern in addition to a few minor points:

Major:

Authors presented a significant experimental data for rx3 gene. I think that there is a technical issue on the genome editing of rx3. The eye less phenotype of the rx3 CRISPR is striking (Fig 3d) but authors must be careful. It is well known that eye development is most susceptible to the injection of excess of nucleotides (morpholino, RNA etc). For example, sense-oligos (morpholino) can induce regressed eye due to the toxic side-effect of 'over-whelming nucleotide' at the embryonic development. Authors used the non-injected surface fish as control, and there was no statistical test by using N=5 CRISPR. To full-fill these points, I request the authors for the statistical test between the eye sizes of surface fish with control CRISPRs and rx3 CRISPR, which are injected with ~25pg of gRNA and ~150pg of Cas9 mRNA. Control CRISPR can be oca2, one of the coauthor have done before (Klaassen et al 2018 Dev Biol).

Minor:

1) Author listed many candidate genes in QTL intervals. Please provide justification of why these candidate genes were 'candidate' Is it because of the former knowledge of genes or gene-network analysis?

2) As for Supplementary Fig 1 that showed the surface fish cross authors used for genome sequence, please explain the reason why this complex cross was made for surface fish.

3) Please describe the parameter to detect the deletion by Manta and LUMPY, ex. cut-off nucleotide number.

Reviewer #1 (Remarks to the Author):

The manuscript “A chromosome level genome of *Astyanax mexicanus* surface fish for comparing population specific genetic differences contributing to trait evolution.” By Warren et al describes a substantially improved assembly, which is used to resolve several biologically interesting phenomena in the species. This is an excellent piece of work, and I only have a couple of comments related to the clarity and consistency of presentation of the genome and QTL mapping data.

1) The scheme used for naming the chromosomal scaffolds. It is not clear what criteria were used to name the scaffolds. Under standard nomenclature chromosome 1 would correspond to the longest chromosome in the karyotype, chromosome 2 the second longest and so forth. However, I did not see where the assembly was anchored to physically defined chromosome although it is clearly anchored to linkage groups. Given the data the authors appear to have on hand the scaffolds might be more appropriately named LG1, LG2 ... if they are ordered based on linkage distance or scaff1, scaff2 ... if they are ordered based on size.

We recognize our chromosome numbering schema was not the traditional approach of assigning by assembled size, but our community voted to assign chromosome numbers based on alignment to numbered genetic linkage groups. This was done to allow easy transition from QTL studies to searches for causative genes. We note this strategy has been done for other species, eg. Domestic cat. We aligned the genetic linkage map sequence markers as described in Gross et al. 2008. using blast to the chromosome level scaffolds (ordered by the Bionano map) and assign each to numbered linkage group, for example scaffolds that align LG1 markers are ordered and oriented and then labelled Chr1. As the reviewer noted, in Fig. 1b we show the overall ordered scaffolds (chromosome) synteny is well conserved across all assigned linkage groups.

2) Unifying the linkage analyses with this nomenclature will also make the presentation of the results more straight-forward. In some cases this seems to have been done, but in others (e.g. in Figure 2 where the same genomic interval appears to be referred to as LG3, LG21 and chr13, and similarly in Figure 4). It would seem that these analyses would be clearer if they were all performed/presented using the same underlying linear scaffold (chr13, or an appropriately named version of this same scaffold).

Yes, we entirely agree. This is actually one of the advantages of the new genome that from now on the new nomenclature can be used. In the case of Figure 2 for example, we can now refer to chr13 and no longer rely on previous (more or less arbitrary) nomenclatures from previous QTL studies. We have changed the text in the figure legend to better highlight this point.

3) A supplemental table showing the correspondence between their chromosomal genomic scaffolds, previous (relevant) linkage groups and other previous or parallel (e.g. NCBI accession numbers) assemblies might also be useful given the history of this

system

We were not able to include a meaningful comparative synteny map of the previous cavefish genome assembly (Astyanax mexicanus 1.0.2) to the current reference (Astyanax mexicanus 2.0) given the low level of scaffold contiguity, 10,735 (no chromosome) vs 2,415 total scaffolds (current chromosomes assembly). However, we take advantage of the NCBI assembly to assembly remapping tool and now provide a spreadsheet that includes the coordinates where all scaffolds from Astyanax mexicanus 1.0.2 are aligned to the chromosomes of Astyanax mexicanus 2.0. This spreadsheet is noted in the methods now under "Defining syntenic regions between cave and surface genomes" section as supplementary data 2 and should aid those interested in transitioning between genetic studies in cave and surface fish. We share the reviewers' interest in seeing a future chromosomes comparison between forms. In fact, we are actively working toward a higher quality cavefish assembly now that allows for a higher resolution synteny comparison between surface and cavefish genomes. Specifically, we are interested in the detailed structural/syntenic changes between these genetically adapted forms of the Astyanax mexicanus species.

Reviewer #3 (Remarks to the Author):

This manuscript reported the first chromosome-level genome assembly of the emerging evolutionary and medical model of the Mexican tetra, *Astyanax mexicanus*. Authors build this genome assembly based on the surface population of *A. mexicanus*, and re-sequenced 4 of cave populations. By utilizing it, authors addressed deletion or loss of the genes in the derived cave populations comparing with ancestral-type surface population and zebrafish. From the view of the full genome, they retested the albinism gene (*Oca2*), and newly tested *Rx3* for eye degeneration. Also, the genetic and genomic bases on cavefish traits (other candidate genes for eye degeneration, insomnia, metabolism, and circadian clock) were surveyed.

Many of these results were novel and a few were reanalyses (*Oca2*, Protas 2005; candidate eye genes, McGough 2013). Their results advance the knowledge of the complex evolutionary processes for cave adaptation. These results will impact cave ecology and biology and in evolutionary biology of rapid adaptation.

Their logic was clear and presented in a proper way.

Because cavefish is an established laboratory model which has ancestral-morph, and polarity of evolution (from surface-type ancestor to cave-morph) and major selection pressures were well defined, the genetics bases for morphological, metabolic and behavioral changes provided in this paper are, I expect, influential to the cave biology and evolutionary biology fields.

There is one major concern in addition to a few minor points:

Major:

Authors presented a significant experimental data for *rx3* gene. I think that there is a

technical issue on the genome editing of *rx3*. The eye less phenotype of the *rx3* CRISPa^{nt} is striking (Fig 3d) but authors must be careful. It is well known that eye development is most susceptible to the injection of excess of nucleotides (morpholino, RNA etc). For example, sense-oligos (morpholino) can induce regressed eye due to the toxic side-effect of 'over-whelming nucleotide' at the embryonic development. Authors used the non-injected surface fish as control, and there was no statistical test by using N=5 CRISPa^{nt}. To full-fill these points, I request the authors for the statistical test between the eye sizes of surface fish with control CRISPa^{nt}s and *rx3* CRISPa^{nt}, which are injected with ~25pg of gRNA and ~150pg of Cas9 mRNA. Control CRISPa^{nt} can be *oca2*, one of the coauthor have done before (Klaassen et al 2018 Dev Biol).

We currently have adult fish injected with gRNAs targeting 6 different genes in the laboratory, and none of these fish show an eye less phenotype. These are unpublished lines and we rather not add them to this manuscript. However, we have added them as a response for the reviewers' benefit:

Gene of interest	Number of adult fish	Eye phenotype
Aanat2	4	no
Rorca	9	no
Hctr2	15	no
NPY	1	no
prph2b	2	no
Oca2	1	no

To more specifically address RNA toxicity, we have performed an additional analysis in larval fish, and this data is now presented in Supplementary Figure 9. We compare *rx3* CRISPa^{nt}s, Cas9 alone controls, and uninjected embryos. We found no significant differences between Cas9 alone and uninjected embryos, whereas eye size is reduced to absent in the majority of the *rx3* CRISPa^{nt} fish. We have performed statistical analysis on these, and show that this effect is significant.

Minor:

1) Author listed many candidate genes in QTL intervals. Please provide justification of why these candidate genes were 'candidate' Is it because of the former knowledge of genes or gene-network analysis?

Genes were referred to as candidates because they are within the QTL confidence interval. Even if a gene within the interval does not have previously known functions contributing to the trait of interest, it does not eliminate it as a potential candidate. We have added this explanation to the methods (Line 682-684).

2) As for Supplementary Fig 1 that showed the surface fish cross authors used for

genome sequence, please explain the reason why this complex cross was made for surface fish.

In order to generate enough high-quality DNA from a single individual, we needed a large female fish. This was the largest available fish at that time in our facilities. In addition, we hoped that by using a cross between different populations we would be able to catch a larger range of genetic variation. However, in retrospect we realized that the increased genetic variation may have rather complicated our alignment and subsequent analysis. For future studies we would recommend to stick to a more inbred individual from a single location. We have added a sentence to the methods to explain the use of this individual.

3) Please describe the parameter to detect the deletion by Manta and LUMPY, ex. cut-off nucleotide number.

We provide in the methods section that we ran both SV callers with default parameter settings and called all SVs without nucleotide size restrictions then used a cutoff of 500bp to 100kb to retain only deletion SVs. We provided a link to our workflows here: <https://github.com/esrice/workflows>. The only parameter for merging deletions is reciprocal overlap, which is specified in the methods sections. We are happy to provide further details if needed.

We made a few additional changes to the manuscript that were not requested by the reviewers but needed clarification:

We added middle initials to two of the authors names.

We corrected the position of the first and last name of one author.

Line 115/116 we changed the wording slightly to make the sentence easier to understand (changes are highlighted in yellow).

We added the whole mount *insitu* protocol to the methods.

We changed the x-axis in Figure 5A from “cave populations” to “populations”.

We added one missing reference (38).

REVIEWER COMMENTS

Reviewer #1 (Remarks to the Author):

I am satisfied with the replies to my initial review. I still disagree philosophically with the departure from standard cytogenetic-based naming scheme for the chromosomes, however the broader community impacts related to uncertainty in renaming chromosomes might outweigh the advantages of using a more traditional naming scheme. Perhaps though, it is best if the community revisits this issue in the future. Altogether, the naming scheme does not impact that major findings reported in this paper and other edits have clarified the relationship between older linkage studies and the current map, at least where they are relevant to topics discussed in the manuscript.

Reviewer #3 (Remarks to the Author):

I applaud that the authors made significant efforts and improved their manuscript by responding to reviewers' comments.

I read it through from the beginning to the end again, and I have no issue with and strongly support the publication of their manuscript in Nat Comm.

Reviewer #4 (Remarks to the Author):

The present study reported a new surface fish of *Astyanax mexicanus* genome, and carried out comparative, functional, developmental and genetic studies of skin color, eye size and hypocretin signaling. The revised manuscript has been improved. However, there are still some issues need to be addressed:

1. Line 145: The NCBI and Ensembl pipelines annotated different numbers of genes. Supplemental data showing the correspondence between the annotations might also be useful. In some cases, annotated ORF of the same gene might be different. Are there any criteria to choose the correct annotation? There is increased number of annotated lncRNA compared to *Astyanax mexicanus* 1.0.2 reference. Do these lncRNAs function in surface fish adaptations?
2. Data of both the Pachón and Molino populations were re-analyzed, which confirmed the mutations of *oca2* are the causal of albinism in cavefish. However, all findings have been reported in previous studies. Thus, it is better to include Figure 2 in the supplementary data. It would be clearer if the Figures were described in order in the main text and figure legend.
3. Line 243-244: Generating engineered CRISPR mutants in *oca2* exon 24 will be helpful to confirm these results.
4. There are many genes harboring in the listed QTL intervals from 8.7-23.0Mb on LG3 and 11.6-18.0 on LG 20. It is still not clear why *rx3*, *opsins*, *opo*, *dusp26* and circadian rhythmicity genes were selected as candidate genes for eye size and eye regression?
5. QTL analysis of eye size in "Surface fish genome reveals new candidate genes from prior QTL studies" section should be combined with the section of Genetic mapping with surface fish genome reveals new candidate genes for eye regression (Line 328).
6. There are quite a few speculations in the results section, which would likely be better suited to the discussion section.
7. Line 201-202: a reference should be included.

Response to Reviewer comments:

Reviewer #1 (Remarks to the Author):

I am satisfied with the replies to my initial review. I still disagree philosophically with the departure from standard cytogenetic-based naming scheme for the chromosomes, however the broader community impacts related to uncertainty in renaming chromosomes might outweigh the advantages of using a more traditional naming scheme. Perhaps though, it is best if the community revisits this issue in the future. Altogether, the naming scheme does not impact that major findings reported in this paper and other edits have clarified the relationship between older linkage studies and the current map, at least where they are relevant to topics discussed in the manuscript.

We agree that our chromosomes should be numbered as noted by the reviewer, that is standard cytogenetic-based naming scheme for the chromosomes. In future genome reference builds, we plan to follow this more standard process after consultation with the community regarding the importance of chromosome nomenclature consistency.

Reviewer #3 (Remarks to the Author):

I applaud that the authors made significant efforts and improved their manuscript by responding to reviewers' comments. I read it through from the beginning to the end again, and I have no issue with and strongly support the publication of their manuscript in Nat Comm.

We thank the reviewer for their time and support of the manuscript.

Reviewer #4 (Remarks to the Author):

The present study reported a new surface fish of *Astyanax mexicanus* genome, and carried out comparative, functional, developmental and genetic studies of skin color, eye size and hypocretin signaling. The revised manuscript has been improved. However, there are still some issues need to be addressed:

*1. Line 145: The NCBI and Ensembl pipelines annotated different numbers of genes. Supplemental data showing the correspondence between the annotations might also be useful. In some cases, annotated ORF of the same gene might be different. Are there any criteria to choose the correct annotation? There is increased number of annotated lncRNA compared to *Astyanax mexicanus* 1.0.2 reference. Do these lncRNAs function in surface fish adaptations?*

In our revisions we offered a spreadsheet resource that allows interested individuals the ability to review the corresponding NCBI sequence coordinates for protein-coding gene model

predictions in the *Astyanax mexicanus* 1.0.2 and 2.0 references. The reviewer also points out a consistent issue of ‘what gene model is best’ for all species that have corresponding gene predictions from the NCBI and Ensembl pipelines. As to the accuracy of independently derived gene models, we contend that without large-scale comparative gene alignments of both sets, NCBI and Ensembl, and a careful manual annotation of observed discrepancies we are not able to provide an acceptable accuracy score for each of the ~25,000 protein-coding genes. The necessary resources to provide such a database of objective accuracy scores for each gene model are not available at this time but we agree nonetheless important to consider.

As we report, the quality of the *Astyanax mexicanus* 1.0.2 reference is the most likely explanation of the higher count of lncRNAs in the surface fish *Astyanax mexicanus* 1.0.2 reference. The reviewer asks a great question about the role of these lncRNAs in cavefish adaptations that is beyond the scope of this study. We are working now to improve the cavefish reference in order to offer the community the opportunity to conduct future studies on how lncRNA may contribute to trait adaptation.

2. Data of both the Pachón and Molino populations were re-analyzed, which confirmed the mutations of oca2 are the causal of albinism in cavefish. However, all findings have been reported in previous studies. Thus, it is better to include Figure 2 in the supplementary data. It would be clearer if the Figures were described in order in the main text and figure legend.

According to the reviewer’s suggestion we have moved Figure 2 to the supplement (now Supplemental Fig. 5) and have reordered the figure panels to represent the correct order in the text. In addition, we have moved Table 1 to the supplement (now Supplementary Table 4), as the Table refers to Figure 2.

3. Line 243-244: Generating engineered CRISPR mutants in oca2 exon 24 will be helpful to confirm these results.

We agree with the reviewer that having engineered CRISPR mutants for *oca2* exon 24 would be a nice addition. Unfortunately, these experiments are beyond the scope of this publication. However, we have previously performed one experiment that, at least in part, addresses the reviewer’s concern. In Klaassen et al. 2018, we have performed complementation experiments with our mutant surface fish (indel in *oca2* exon 21) and Pachón cavefish (which carry the exon 24 deletion). We observed that transheterozygotes with our engineered allele and the Pachón cave allele do not show complementation of the pigmentation phenotype, strongly suggesting that a deletion of *oca2* exon 24 is causing albinism.

4. There are many genes harboring in the listed QTL intervals from 8.7-23.0Mb on LG3 and 11.6-18.0 on LG 20. It is still not clear why rx3, opsins, opo, dusp26 and circadian rhythmicity genes were selected as candidate genes for eye size and eye regression?

We apologize for the confusion. These candidate genes were not necessarily chosen because of a relation to eye size, but more generally as putative candidate genes related to locomotor activity differences for which this QTL analysis was performed. We believe the confusion stems from a few sentences that refer to eye regression earlier in this paragraph, that were placed wrongly in this section. We have deleted these sentences now (see also answer to comment #5).

5. QTL analysis of eye size in "Surface fish genome reveals new candidate genes from prior QTL studies" section should be combined with the section of Genetic mapping with surface fish genome reveals new candidate genes for eye regression (Line 328).

We have addressed the comment by deleting sentences related to eye development from the first section. These were wrongly placed, and we thank the reviewer for pointing out this inconsistency.

6. There are quite a few speculations in the results section, which would likely be better suited to the discussion section.

We have carefully read through the results again and removed speculations or highlighted limitations of the statements.

7. Line 201-202: a reference should be included.

We reference Klaassen et al., 2018 at this position.